# *Enterococcus* and COVID-19: The Emergence of a Perfect Storm?

**Dan Alexandru Toc** [1,*] , **Razvan Marian Mihaila** [2] , **Alexandru Botan** [1,*] , **Carina Nicoleta Bobohalma** [1] , **Giulia Andreea Risteiu** [1] , **Bogdan Nicolae Simut-Cacuci** [1] , **Bianca Steorobelea** [1] , **Stefan Troanca** [1] and **Lia Monica Junie** [1]

1   Department of Microbiology, Iuliu Hatieganu University of Medicine and Pharmacy,
    400012 Cluj-Napoca, Romania; bobohalma.carina.nico@elearn.umfcluj.ro (C.N.B.);
    risteiu.giulia.andreea@elearn.umfcluj.ro (G.A.R.); simut.cacuci.bogdan.nicolae@elearn.umfcluj.ro (B.N.S.-C.);
    steorobelea.bianca@elearn.umfcluj.ro (B.S.); troanca.stefan@elearn.umfcluj.ro (S.T.);
    junie.lia.monica@elearn.umfcluj.ro (L.M.J.)
2   Centre Hospitalier Régional d'Orléans, 14 Av. de l'Hôpital, 45100 Orléans, France;
    mihaila.razvan@elearn.umfcluj.ro
*   Correspondence: toc.dan.alexandru@elearn.umfcluj.ro (D.A.T.); botan.alexandru@elearn.umfcluj.ro (A.B.)

**Abstract:** (1) Background: Based on the uncontrolled use of antibiotics and the lack of worldwide-accepted healthcare policies, the COVID-19 pandemic has provided the best premises for the emergence of life-threatening infections. Based on changes described in the intestinal microbiome, showing an increased number of *Enterococcus* bacteria and increased intestinal permeability due to viral infection, infections with *Enterococcus* have taken the spotlight in the healthcare setting; (2) Methods: We conducted a brief review in order to analyze the relationship between the two pathogens: the SARS-CoV-2 virus and the *Enterococcus* bacterial genus. We searched in PubMed, the Cochrane Library electronic database and MedNar and included twenty-one studies based on relevance; (3) Results: The existing studies show a statistically significant difference in the composition of the intestinal microbiome, favoring *Enterococcus* genus, when compared to a control group. Changes also seem to persist over a period of time, suggesting possible implications for long COVID. Regarding bloodstream infections, *Enterococcus* is statistically significantly isolated more often when compared to the pre-COVID-19 era, and to a control group of non-COVID-19 patients. (4) Conclusions: The intimate synergy between COVID-19 and *Enterococcus* has the potential to pose a real threat to human healthcare, and more extensive research is needed to explore the relationship between these two pathogens.

**Keywords:** *Enterococcus*; VRE; COVID-19; long COVID; SARS-CoV-2

## 1. Introduction

The COVID-19 pandemic has taken the world on a race against the clock for providing the best diagnostic tools, proper treatment and the implementation of appropriate prevention measures. The implications of this pandemic from a socioeconomic and scientific point of view cannot be easily comprehended, since there is a continuous analysis of its aftermath. Since many aspects are still lacking proper understanding, research in this field continues to provide much-needed information in order to bridge the existing gaps [1–3].

The *Enterococcus* genus provides some of the most dangerous bacterial species involved in human infections, such as *Enterococcus faecium* (vancomycin-resistant), or the latest, *Enterococcus faecium* (linezolid-resistant) [4–6]. In the past five years, many efforts were executed worldwide in order to develop new drugs able to cure this type of infection or to develop new strategies meant to keep the trend of increased antimicrobial resistance under control. Beside the most notorious species of the genus, such as *Enterococcus faecalis* and *Enterococcus faecium*, of notable importance are *Enterococcus gallinarum* and *Enterococcus caseliflavus*, which are intrinsic resistant to glycopeptides, due to the *vanC* gene, but also

*Enterococcus raffinosus,* which has proven to be able to harbor the dangerous *vanA* gene. New insights in *non-faecalis non-faecium enterococcus* have shown that almost every species of this genus has the potential to pose a real threat for human health, and continuous research on this niche is necessary [7–11].

An unexpected aftermath of the COVID-19 pandemic is the high number of cases of *Enterococcus* diagnosed in patients with COVID-19. Many hypotheses have arisen for these findings, but since the physiopathology of the infection with SARS-CoV-2 remains incompletely described, no definitive explanation is available [12,13]. In this paper we aim to untangle the complex interrelation between viral infection with SARS-CoV-2 and the bacterial genus *Enterococcus* by providing a brief review of the existing medical literature.

## 2. Materials and Methods

In order to evaluate the relationship between *Enterococcus* and COVID-19, we performed a brief review of the existing literature. We searched relevant articles on PubMed, the Cochrane Library electronic database and MedNar, up to the 1st of September 2021.

We considered the following terms included in the studies' titles or abstracts: "*enterococcus*" combined with the operator "AND" along with "COVID-19", "SARS-CoV-2", "COVID-19 infection" or "SARS-CoV-2 infection". The search was performed by two individual researchers and the results were confronted afterwards. We excluded duplicates and studies written in languages other than English, French or Spanish.

Figure 1 presents the flow diagram of the search we performed based on the criteria describes above, as well as the number of articles included, and their distribution based on selected criteria.

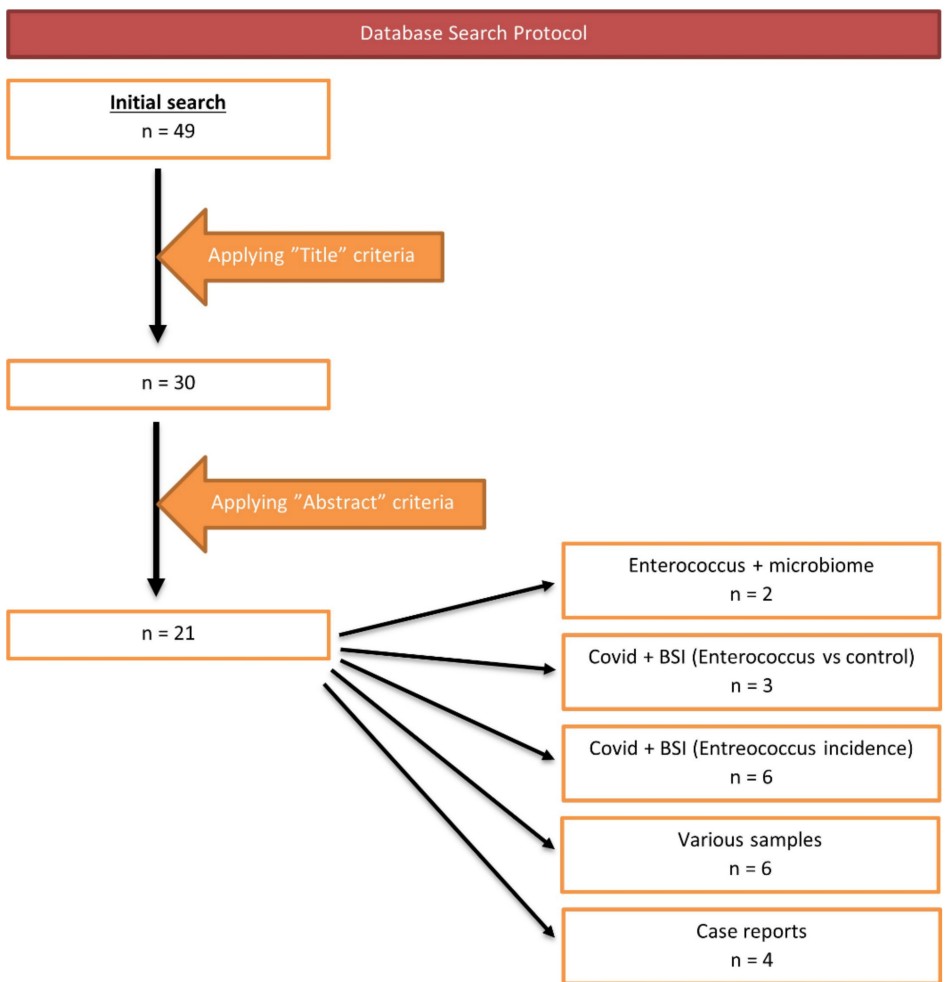

**Figure 1.** Flow diagram of the database search protocol.

## 3. Results

Based on the protocol described above, we included twenty-one studies in our review. We organized them as follows: *Enterococcus* and bloodstream infections (BSIs), *Enterococcus* from various samples from COVID-19 patients, *Enterococcus* and gut microbiota (GM) in COVID-19 patients and *Enterococcus* and COVID-19 case reports.

Table 1 presents the articles included in our article that involved the study of microbiome in patients with COVID-19. Both studies, published in 2021, used stool samples to describe the changes in the intestinal microbiome that were present in patients with COVID-19.

**Table 1.** Intestinal microbiome.

| No. | Author | Year | Country | Sample | Diagnostic Technique | COVID-19 Patients | Additional Information |
|---|---|---|---|---|---|---|---|
| 1 | Gaibani, P. et al. [14] | 2021 | Italy | Stool sample | Illumina MiSeq | 69 | The GM of COVID-19 patients showed the enrichment of known or potential opportunistic pathogens, such as *Enterococcus*, *Staphylococcus*, *Serratia* and *Collinsella* (*p* value ≤ 0.02) |
| 2 | Zhou, Y. et al. [15] | 2021 | Republic of China | Stool sample | MagPure Stool DNA KF kit B | 127 | *Saccharomyces* and *Enterococcus* were significantly enriched in patients with fever |

Tables 2 and 3 contain the articles that describe BSIs involving *Enterococcus* in patients with COVID-19. Based on the study design, the control group is usually composed of patients without COVID-19. The diagnostic tools that were used in these studies were different, showing a heterogenous availability of proficient diagnostic tools. *Enterococcus faecalis* seems to be the specie of the genus *Enterococcus* most commonly isolated from patients with COVID-19. Studies that described the incidence of *Enterococcus* in BSIs in patients with COVID-19 showed a variable outcome. Vancomycin-resistant *Enterococcus* (VRE) remains a constant threat, being described in most included articles.

**Table 2.** BSI (COVID vs. control).

| No. | Author | Year | Country | Diagnostic Technique | COVID-19 Patients | Non-COVID-19 Patients | *Enterococcus* Spp. of COVID-19 + | *Enterococcus* Spp. of COVID-19 − | *E. Faecium* | *E. Faecalis* |
|---|---|---|---|---|---|---|---|---|---|---|
| 1 | Hughes, S. et al. [16] | 2020 | UK | MALDI-TOF | 836 | 216 | 1 (0.47%) | - | - | - |
| 2 | DeVoe, C. et al. [17] | 2021 | USA | MiSeq | 314 | 14,332 | 8 (2.6%) | 48 (0.3%) | 2 (0.6%) | 6 (1.9%) |
| 3 | Cuntrò, M. et al. [18] | 2021 | Italy | VITEK2 | 1911 | - | 106 (5.54%) | 56 (2.93%) | 32 (1.67%) | 74 (3.78%) |

*Enterococcus* was also described from other, different samples. Table 4 presents the studies included in our review that evaluate *E. faecium* and *E. faecalis* from all the samples.

Table 5 presents the case reports involving the isolation of *Enterococcus* from different samples from patients with COVID-19. *E. faecalis* was the species involved in most case reports. The sites of the infections described were representative for most of the common sites of infections for *Enterococcus*: endocarditis, pneumonia, etc.

**Table 3.** BSI (*Enterococcus* incidence).

| No. | Author | Year | Country | Diagnostic Technique | COVID-19 Patients | *Enterococcus* Spp. of COVID-19 + | *E. Faecium* | *E. Faecalis* | VRE | Other Resistances |
|---|---|---|---|---|---|---|---|---|---|---|
| 1 | Giacobbe, D.R. et al. [19] | 2020 | Italy | VITEK-2 | 78 | 12/45 BSI (26.6%) | 4 (8.8%) | 8 (17.7%) | 1 VRE (*E. faecium*) | 4/4 *E. faecium* were ampicillin-resistant (100%) |
| 2 | Palanisamy, N. et al. [20] | 2021 | India | VITEK-2 | 750 | 11/64 BSI (17.2%) | - | - | 2 VRE | 81.8% of *Enterococci* were MDRO. Ampicillin (81.8%), Ciprofloxacin (81.8%), Tetracycline (54.5%), Erythromycin (90.9%), Teicoplanin (18.1%) |
| 3 | Abelenda-Alonso, G. et al. [21] | 2021 | Spain | - | 100 | 42/169 isolates (24.85%) | 10 (5.91%) | 32 (18.93%) | 1 VRE | - |
| 4 | Posteraro, B. et al. [22] | 2021 | Italy | MALDI-TOF | 293 | 15/58 BSI (20.7%) | 2 (3.44%) | 10 (17.24%) | - | - |
| 5 | Bonazzetti, C. et al. [23] | 2020 | Italy | VITEK MS MALDI-TOF | 89 | 53/93 BSI (55.8%) | 26 (27.95%) | 26 (27.95%) | 5 VRE (*E. faecium*) | - |
| 6 | Signorini, L. et al. [24] | 2021 | Italy | - | 92 | 6/57 BSI (10.5%) | - | - | 3 VRE | - |

**Table 4.** Various samples.

| No. | Author | Year | Country | Sample | Diagnostic Technique | COVID-19 Patients | *E. Faecium* | *E. Faecalis* |
|---|---|---|---|---|---|---|---|---|
| 1 | Kampmeier, S. et al. [25] | 2020 | Germany | Blood culture samples and pleural drainage | MALDI-TOF-MS | 3 | 3 | - |
| 2 | O'Toole, R.F. et al. [26] | 2021 | Spain | Urine | - | 72 | 4 | - |
| | | | Italy | Blood culture | - | 78 | - | 14 |
| 3 | Senok, A. et al. [27] | 2021 | United Arab Emirates | Blood and central-line cultures, endotracheal aspirates and urine | BioFire FilmArray | 29,802 | 10 | 18 |
| 4 | Cultrera, R. et al. [28] | 2021 | Italy | Blood, urine, or respiratory specimens obtained with bronchoalveolar lavage (BAL) or bronchial aspirate (BASP)/BSI | MALDI-TOF by VITEK MS, VITEK 2 | 28 | 10 | 14 |
| 5 | Saeed, N.K. et al. [29] | 2021 | Kingdom of Bahrain | Blood culture, sputum culture, stool culture, endotracheal aspirate or bronchoalveolar lavage culture | MALDI-TOF MS; BD Phoenix | 261 | 24 | 20 |
| 6 | Calderaro, A. et al. [30] | 2021 | Italy | Lower respiratory tract | MALDI-TOF using a VITEK MS instrument | 90 | 3 | 11 |

**Table 5.** Case reports.

| No. | Author | Year | Country | Sample | Diagnostic Technizque | COVID-19 Patients | *E. Faecium* | *E. Faecalis* |
|---|---|---|---|---|---|---|---|---|
| 1 | Amaral, L. et al. [31] | 2020 | Brazil | Nosocomial pneumonia | - | 1 | - | 1 |
| 2 | Ramos-Martínez, A. et al. [32] | 2020 | Spain | Blood or valve culture | - | 2 | - | 2 |
| 3 | Serrano, O.K. et al. [33] | 2020 | USA | Perinephric collection | - | 1 | 1 vancomycin-resistant *Enterococcus* | |
| 4 | Sanders, D.J. et al. [34] | 2020 | USA | Aortic valve culture | - | 1 | - | 1 |

## 4. Discussion

The COVID-19 pandemic has launched the best premises for the development of highly resistant bacterial strains due to unregulated antimicrobial use and the lack of proper worldwide-accepted protocols. *Enterococcus* genus represents one of the most common findings in human infections. It is no surprise that during the pandemic, a high number of this type of infection was anticipated. However, it is unclear why the number of *Enterococcus* is so high in COVID-19 patients. The relationship between *Enterococcus* and the SARS-CoV-2 virus is also unclear.

We wanted to tackle this issue by analyzing the existing articles in a brief review format. Based on the included articles, we tackled the issue of *Enterococcus* infections in the COVID-19 era in a step-by-step approach. We initially focused on the BSIs (bloodstream infections) involving *Enterococcus* from an incidence and outcome point of view without ignoring other types of samples. We then compared the findings from microbiome studies. Lastly, we analyzed the existing case reports involving these two pathogens.

### 4.1. Enterococcus and Bloodstream Infections

Most of the articles in this brief review described *Enterococci* solely in the context of BSI. In order to better understand the relationship between COVID-19 and BSIs with *Enterococcus,* we further classified them into articles that compared their findings with a control group (such as patients with influenza, COVID-19-negative patients or a different time period) and articles that simply described the incidence of *Enterococcus* infections among COVID-19 patients that developed a BSI [16–24].

In the first group, one study compared the number of *Enterococcus* cases within a COVID-19-positive group with the number of cases within an influenza-positive group [16]; another one used two control groups, one consisting of influenza patients and one consisting of just COVID-19-negatives [17]; a third one made a comparison between the incidence of *Enterococcus* BSI during similar periods, 1 year apart, before and after the start of the pandemic (2019 and 2020) [18]; and the fourth study compared the proportions of BSIs caused by *Enterococcus* during three different years [23].

No *Enterococcus* isolates were found from either group of influenza patients from the two studies that analyzed them, so no statistical analysis comparing them to COVID-19 groups could be made. The incidence of *Enterococcus* BSI among COVID-19-positive patients varied between 0.47% and 2.6% [16,17].

All the BSI studies that used control groups other than influenza found a solid association between *Enterococcus* BSI and COVID-19. De Voe at al. describe that *Enterococcus* spp. BSI occurred in 2.6% of COVID-19 patients (8/314 cases—6 *E. faecalis* and 2 *E. faecium*), compared to the 0.33% of the COVID-19-negative group (48/14,332 cases), the adjusted odds ratio being 3.75 (95% CI: 1.49–9.41) [17]. Cuntrò et al. compared the incidence of *E. faecalis* and *E. faecium* BSIs in an ICU before the pandemic (Feb 22nd to May 21$^{st}$, 2019) with their incidence during the pandemic (Feb 22nd to May 21$^{st}$, 2020). What they determined was that there was a substantial increase in the number of *E. faecalis* cases—28 in 2019 vs. 74 in 2020 ($p < 0.001$, Fisher's test), but no significant increase for *E. faecium*—27 in 2019 vs. 32 in 2020 ($p = 0.41$, Fisher's test) [18]. Bonazzetti et al. compared the proportions of BSIs caused by *Enterococcus* between 2020, 2019 and 2018. The rate was significantly higher in 2020 (71.7% vs. 33.3% and 20%; $p = 0.016$) [23].

The rest of the studies included in this brief review, involving BSI with *Enterococcus,* mainly provide an incidence of this infection that varies between 17.2% and 37.5% [19,20,22,24]. Giacobbe et al. and Posteraro et al. went one step further, and described the incidence of the most common species, *E. faecium* and *E. faecalis* [19,22]. In those studies, the incidence of *E. faecium* varied between 3.5% and 8.88%, while *E. faecalis* remained somehow constant at around 17%. Therefore, we can notice that there could be a tendency for a greater occurrence of *E. faecalis* cases in COVID-19 patients, rather than *E. faecium*, which would be consistent with the other earlier findings described by Bonazzetti et al. [23].

Vancomycin-resistant *Enterococci* were also present within these studies. Most of them report the frequency between 2.38% and 18.1% of all *Enterococci* [19–21]. However, Signorini et al. reported a much higher incidence of VRE − 33.3% [24].

Palanisamy et al. described that 81.8% of *Enterococci* was MDR: resistance to erythromycin was reported in 90.9% of isolates, to ampicillin and ciprofloxacin in 81.8%, to tetracycline in 54.5% and to teicoplanin in 18.1%. In addition, another relevant observation from this study was the high mortality [20].

As presented so far, *Enterococcus* seems to be isolated from BSI at much higher rate than before the pandemic, and mortality in cases that involve COVID-19 and *Enterococcus* BSI is high. Additionally, antimicrobial resistance appears to be an additional threat in this situation. Further analysis is required in order to provide an adequate conclusion.

### 4.2. Enterococcus from Various Samples from COVID-19 Patients

Six of the studies included in this review analyzed secondary infections in patients diagnosed with COVID-19. It was observed that a wide range of bacterial infections occurred in these patients, notably: *Streptococcus pneumoniae, Staphylococcus aureus, Staphylococcus epidermidis, Pseudomonas aeruginosa, Escherichia coli, Klebsiella pneumoniae, Enterococcus faecium, Enterococcus faecalis, Acinetobacter baumannii* or *Haemophilus influenzae*.

Two of the articles emphasized finding *Enterococcus* not only in samples recovered from patients, but in environmental samples too. In one of them, after detecting five cases of COVID-19 with VRE infections, some tests from environmental sites were made. Eleven cases of VRE were determined from those samples. *E. faecium* was the bacteria isolated, and further analysis described two clusters of closely related strains. Interestingly, one of these clusters corresponded to COVID-19-positive patients. The other study revealed that in the USA, *E. faecalis* and *E. faecium* have emerged as further common nosocomial infections after MRSA, being responsible for 7.4% and 3.7% of all HAIs. The researchers detected isolates of VRE in environmental samples, highlighting what an important role contaminated surfaces have in VRE transmission to COVID-19 patients [25,26].

Serano et al. outlined the descending order of the specimen types depending on the number of positive cultures in secondary infection: blood, endotracheal aspirate, urine, sputum wound swab or bronchoalveolar lavage. Regarding *E. faecalis* and *E. faecium*, these were in the first half of frequency scale from positive cultures [33].

Furthermore, Saeed N et al. have shown that these two bacteria occurred in 44 cases of 261 patients with secondary co-infections. The pathological products that were used to find the microorganisms were blood culture, sputum culture, stool culture, endotracheal aspirate and bronchoalveolar lavage culture [29].

*Enterococcus* has been isolated from a wide variety of pathological products from patients with COVID-19 and, thus, implies an eminent threat for human health. It also raises awareness of the importance of microbiologic diagnosis in order to provide long-term proper healthcare.

### 4.3. Enterococcus and Gut Microbiota of COVID-19 Patients

Two of the articles focused on the gut microbiota (GM) changes that occurred in patients diagnosed with COVID-19 from Italy and China. Gaibani et al. analyzed stool samples from patients with COVID-19 from three different hospitals in Bologna (Italy), where the results from the next-generation sequencing method were compared with the publicly available sequences from matching sex and age in healthy Italians and critically ill non-COVID-19 patients [14]. Zhou et al. analyzed samples from Wuhan Union Hospital, where the GM of two groups of moderate COVID-19 patients were compared, one of the groups presenting fever ($\geq$37.3 °C) and the other one without fever. Compared to the data prior to the pandemic, an increased number of secondary infections was reported, which led to monitoring the GM of COVID-19 patients. It was decided that it has an important effect on mediating the inflammatory response, favoring the production of pro-inflammatory cytokines, such as IL-6, thus causing fever in virus-infected patients [15].

In both articles, following examination of samples from COVID-19 patients, a disruption to the microbiota homeostasis was observed. Particular to this infection was the reduced diversity of the microbiota (alpha diversity) in comparison to the healthy controls (*p* value = 0.0008, Wilcoxon test) [14)], with reduced health-associated microorganisms from the *Ruminococcaceae, Bacteroidaceae* and *Lachnospiraceae* families, responsible for producing short-chain fatty acids (SCFA), important in human immunological and metabolic homeostasis, featured by an increasing growth of potential pathogens, especially *Enterococcus*, in addition to *Staphylococcus, Lactobacillus* and *Serratia* (*p* value $\leq$ 0.02) [14]. There seems to be a causal relationship between *Enterococcus* spp. and bloodstream infections (BSIs) developed in COVID-19 patients. Two of the dominant species responsible were *E. faecalis* (1.8%) and *E. faecium* (8.4%) [14].

An interesting observation was made when analyzing the GM profiles of COVID-19 patients. Those who entered the ICU and developed BSI presented a loss of alpha diversity (*p* value $\leq$ 0.004) accompanied by an abundance of *Enterococcus* (*p* value $\leq$ 0.001), compared to the opposite profiles, who did not enter the ICU and did not develop BSI [14]. A similar depletion in alpha diversity was observed in patients presenting fever, in contrast to those without fever. Moreover, fever could be associated with a decreased relative abundance of *Bacteroidetes*, as well as a significant enrichment of *Enterococcus faecalis, Saccharomyces cerevisiae* and *Haemophilus parainfuenzae*, whereas the GM of non-fever patients indicated the growth of *Anaerostipes*, a butyrate-producing bacteria which suppresses the inflammatory cytokine production. The latter has been observed to be lacking in those with fever [15]. Furthermore, the GM of critically ill patients who tested negative for COVID-19 showed the presence of *Enterobacteriaceae* (*Klebsiella* spp.) (*p* value $\leq$ 0.001), which mainly distinguished the samples from patients who tested positive for the infection [14].

Due to the ability of SARS-CoV-2 to enter the cells of the digestive tract, therefore leading to enteric manifestations through virus-induced immune-mediated damage, the GM of the infected patients developed an accumulation of opportunistic pathogens or pathobionts, potentially antibiotic-resistant [14]. Consequently, the crossing of the bacteria to the circulation through the damaged intestinal epithelium was facilitated, which is very clinically relevant. Finally, studies have found that GM plays a major role in the pathogenesis of inflammation and the evolution of infection with SARS-CoV-2, controlling the immune responses.

*4.4. Enterococcus and COVID-19—Case Reports*

There were reported cases of COVID-19 patients presenting a secondary bacterial infection in four of the articles, which is not atypical in the context of infection with SARS-CoV-2.

In one case, a patient presented with the following symptoms: fever, dyspnea and cough. He was later diagnosed with nosocomial pneumonia caused by *Enterococcus faecalis* and tested positive for COVID-19. It is yet unknown if the pulmonary affection was determined by the bacterial co-infection or by the virus alone [31].

One article outlined a higher incidence of hospital-acquired infected endocarditis (HAIE) between March and April 2020, compared to the correspondent months 5 years prior to the pandemic (*p* = 0.033). Four cases of HAIE were reported in this period, the pathogenesis of two of them being *Enterococcus faecalis*, and for the others, *Staphylococcus aureus* and *Candida albicans*. The diagnosis was made after analyzing blood or valve culture. All four patients with HAIE underwent central venous and urinary catheterization when admitted to the hospital, which appeared to be the source of infection [32]. Co-infections with bacteria and fungi are a common complication in COVID-19 cases [35]. Another article discussed the case of a patient diagnosed with both COVID-19 and infected endocarditis (IE), who presented nonproductive cough, shortness of breath and fatigue. After a blood culture analysis, the etiology of IE was *Enterococcus faecalis*. After a persistent bacteremia following antibiotic therapy with vancomycin and gentamicin and an insertion of a new

catheter, the patient suffered an aortic valve surgery, which identified a vegetation >10 mm of *Enterococcus faecalis*. The treatment changed to ampicillin and ceftriaxone [34].

The last case-report article presents a patient who underwent a simultaneous heart-kidney transplant (SHKT) previous to the first cases of SARS-CoV-2 in the respective area. In the weeks following the intervention, he was admitted to the hospital for respiratory failure, open non-healing wounds and multiple secondary infections due to opportunistic bacteria. A perinephric collection showed vancomycin-resistant *Enterococcus* (VRE) and his blood samples grew out MRSA bacteremia. He was also tested positive for SARS-CoV-2, in spite of being asymptomatic. The patient was immunosuppressed following the procedure, which led to complications [33].

## 5. Conclusions

In the COVID-19 era, a new threat seems to be emerging: the *Enterococcus* genus. From BSI, to changes in the microbiome, a new pathogen-to-pathogen relationship between SARS-CoV-2 and *Enterococcus* seems to take the spotlight. Although the exact mechanism remains unknown, the viral infection seems to cause changes in the bacterial microbiome, favoring *Enterococcus* and increasing intestinal permeability, which provides the perfect circumstances for *Enterococcus* bacteria to develop invasive infections. With a constant number of VRE being reported in these articles, the threat of life-threatening infections seems to be higher than ever [36]. Moreover, special attention should be considered regarding long COVID and changes in the microbiome, which also seem to be linked. With COVID-19's aftermath being constantly evaluated, we might be facing the emergence of a perfect storm, and we should be prepared to tackle it in the best possible way.

**Author Contributions:** Conceptualization, D.A.T. and L.M.J.; methodology, D.A.T. and R.M.M.; software, R.M.M. and A.B.; validation, D.A.T., R.M.M., A.B. and L.M.J.; formal analysis, D.A.T., C.N.B., G.A.R., B.N.S.-C., B.S. and S.T.; investigation, C.N.B., G.A.R., B.N.S.-C., B.S. and S.T.; resources, C.N.B., G.A.R., B.N.S.-C., B.S. and S.T.; data curation, D.A.T. and A.B.; writing—original draft preparation, C.N.B., G.A.R., B.N.S.-C., B.S. and S.T.; writing—review and editing, D.A.T., A.B., R.M.M. and L.M.J.; visualization, A.B.; supervision, L.M.J. All authors have read and agreed to the published version of the manuscript.

**Funding:** This research received no external funding.

**Institutional Review Board Statement:** Not applicable.

**Informed Consent Statement:** Not applicable.

**Data Availability Statement:** Not applicable.

**Conflicts of Interest:** The authors declare no conflict of interest.

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
