# Peer review of "Enterococcus and COVID-19: The Emergence of a Perfect Storm?"

_2673-8937, doi:10.3390/ijtm2020020_

Round 1
Reviewer 1 Report
The manuscript titled “Enterococcus and COVID-19: The emergence of a perfect storm?” states as objective “We conducted a brief review in order to analyze the relationship between the two pathogens: Sars-15 CoV-2 virus and Enterococcus bacterial genus.”
The objective was adequately achieved, the manuscript is very interesting and follows the editorial criteria, it has good scientific rigor and the cited references have been extensively reviewed.
It is important to review the writing of microorganisms throughout the text, following the binomial nomenclature: for example: pp 1, line 43, 44; pp 2 line 52; pp 3 lines 69, 70, 71; pp 4 Title; pp 6 line 127, 129, 132, 140; check all the text carefully. Some author surnames appear in lowercase
In many parts of the text Enterococcus, Enterococcus, Enterococci and Enterococci are used interchangeably, review and unify with the correct nomenclature.
A detailed discussion of the topic is made in the proposed order, however, I would like to try to explain more at the end of the discussion in some paragraph the possible causes of the relationship between the frequency of Enterococcus and Sars-CoV2, especially of those species more resistant to antibiotics and what is the future perspective of this demonstrated increase.
I congratulate the authors for the well-accomplished manuscript.
Author Response
Thank you for this review.
We provide a point-by-point response to the comments.
Comments and Suggestions for Authors
The manuscript titled “Enterococcus and COVID-19: The emergence of a perfect storm?” states as objective “We conducted a brief review in order to analyze the relationship between the two pathogens: Sars-15 CoV-2 virus and Enterococcus bacterial genus.”
The objective was adequately achieved, the manuscript is very interesting and follows the editorial criteria, it has good scientific rigor and the cited references have been extensively reviewed.
It is important to review the writing of microorganisms throughout the text, following the binomial nomenclature: for example: pp 1, line 43, 44; pp 2 line 52; pp 3 lines 69, 70, 71; pp 4 Title; pp 6 line 127, 129, 132, 140; check all the text carefully. Some author surnames appear in lowercase
Thank you for the comment. We made the changes.
In many parts of the text Enterococcus, Enterococcus, Enterococci and Enterococci are used interchangeably, review and unify with the correct nomenclature.
Thank you for the comment. We made the changes.
A detailed discussion of the topic is made in the proposed order, however, I would like to try to explain more at the end of the discussion in some paragraph the possible causes of the relationship between the frequency of Enterococcus and Sars-CoV2, especially of those species more resistant to antibiotics and what is the future perspective of this demonstrated increase.
Thank you for this comment. Although there are several hypotheses that can describe a physio pathologic relationship between the pathogens it would be hazardous to jump to any conclusions at this stage. We mentioned the most relevant – intestinal permeation. Regarding the isolation of VRE we propose that the existing high antimicrobial pressure due to unregulated antimicrobial use may influence this. We added an extra reference regarding this issue.
I congratulate the authors for the well-accomplished manuscript.
Reviewer 2 Report
Many names of microorganisms must be written in italics, such as line 21, line 43 etc
the work is well presented and developed. It is of interest to the scientific community as it is well known that covid 19 has also favored infectious complications of a bacterial nature. the authors underlined the association between enterococci and covid. the work was conducted with methodological rigor, in a clear way, supporting adequate conclusions.
Author Response
Thank yoy for this review.
We provide a point-by-point response.
Comments and Suggestions for Authors
Many names of microorganisms must be written in italics, such as line 21, line 43 etc
Thank you for the comment. We made the changes.
the work is well presented and developed. It is of interest to the scientific community as it is well known that covid 19 has also favored infectious complications of a bacterial nature. the authors underlined the association between enterococci and covid. the work was conducted with methodological rigor, in a clear way, supporting adequate conclusions.
Reviewer 3 Report
This manuscript reviewed the association of COVID-19 with Enterococcus. Some comments should be addressed for improvement.
- PubMed, Cochran library, and Medline are widely used databases. Please provide the reason for choosing Med Nar instead of Medline.
- Please provide the detailed search strategies for each database in revised supplementary table.
- Figure 1: PRISMA guideline should be applied.
Author Response
Thank you this review
We provide a point by point response.
Comments and Suggestions for Authors
This manuscript reviewed the association of COVID-19 with Enterococcus. Some comments should be addressed for improvement.
- PubMed, Cochran library, and Medline are widely used databases. Please provide the reason for choosing Med Nar instead of Medline.
- Please provide the detailed search strategies for each database in revised supplementary table.
- Figure 1: PRISMA guideline should be applied.
Thank for these comments. This paper is a brief review not a systematic review and although we try to maintain some of the rigors of a systematic review, there are not specific guidelines that need to be applied for a brief review. We chose to use Med Nar instead of Medline because it provides a deep web search that may provide more articles to choose from which is very useful when approaching a niche type of subject. PubMed, Cochrane and Medline are somehow similar, and we chose to exclude Medline in favor of Med Nar for a more comprehensive search. We believe that this search provides the best outcome regarding the needed articles for our paper.
Round 2
Reviewer 3 Report
I found that all comments were addressed.